# Correlates of School Children’s Handwashing: A Study in Tibetan Primary Schools

**DOI:** 10.3390/ijerph16173217

**Published:** 2019-09-03

**Authors:** Chang Sun, Qingzhi Wang, Sasmita Poudel Adhikari, Ruixue Ye, Sha Meng, Yuju Wu, Yuping Mao, Hein Raat, Huan Zhou

**Affiliations:** 1Department of Health and Social Behavior Science, West China School of Public Health, Sichuan University, Chengdu 610041, China (C.S.) (Q.W.) (S.P.A.) (R.Y.) (S.M.) (Y.W.); 2Department of Communication Studies, California State University, Long Beach, CA 90802, USA; 3Department of Public Health, Erasmus MC—University Medical Center Rotterdam, 3000 CA Rotterdam, The Netherlands

**Keywords:** handwashing behavior, school children, rural areas, the Tibetan nationality, theory of reasoned action, structural equation model

## Abstract

Hand hygiene, including handwashing by children, has been reported to contribute to the prevention of various infectious conditions. This study aims to explore the correlates of handwashing behavior among 1690 fourth to sixth grade primary school students in 19 Tibetan primary schools (Golog, Qinghai, China). The theory of reasoned action (TRA) was applied. Data was collected by questionnaire. Structural equation modeling (SEM) analysis showed that students’ attitude (β = 0.22, 95% CI 0.13–0.31) and subjective norms in terms of compliance to teachers’, parents’ and peers’ suggestions to wash hands (β = 0.09, 95% CI 0.01–0.18) were directly associated with students’ handwashing behavior. Students’ knowledge (β = 0.04, 95% CI 0.03–0.07) had an indirect association with handwashing behavior, mediated by students’ attitudes and subjective norms. Subjective norms (β = 0.12, 95% CI 0.07–0.17) were also indirectly correlated with handwashing through students’ attitudes. Therefore, our study supported the theory of reasoned action through our findings that students’ attitude and knowledge, and also attitudes from teachers, parents and peers were correlated with student handwashing behavior. Students reported higher level of compliance to teachers than to their parents and classmates. Based on this information, we recommend teacher-involved participatory hygiene education to promote students’ handwashing behaviors in areas at high risk for infectious diseases that can be prevented by handwashing.

## 1. Introduction

Hand hygiene has been recommended as an important way of keeping individuals from diseases such as foodborne parasitic infections [1,2,3]. Systematic reviews of nonexperimental and experimental studies related to handwashing have supported the importance of handwashing as a relevant infection control measure [2,4]. Numerous studies established the associations between handwashing and the prevalence of distinct infections, and these studies supported the causality of these associations by taking into account temporality, strength, plausibility, and consistency of the associations [4]. Therefore, the important role of handwashing and health education was highlighted in promoting adequate handwashing, as a relevant contribution to breaking the transmission cycle of distinct infections (e.g., alveolar echinococcosis, cystic echinococcosis) [5].

Although handwashing is a simple and efficient method for reducing the risk of infectious diseases, studies have shown the relatively low adherence to the recommended handwashing; the reported percentages of adherence vary between 1.8% and 78%, depending on the population, and context [6,7]. A large body of the existing literature on handwashing compliance primarily focuses on health care professionals such as doctors, nurses, and medical school students instead of patients, school children, or other populations at high risk of infectious diseases [8,9].

Studies on the behavioral determinants of adequate hand hygiene remain scarce and lack conclusive results [9]. A few studies primarily applied single-factor and multi-factor regression analysis to investigate the main influencing factors of poor handwashing behavior [10,11]. However, individuals’ behaviors may be influenced by important people or groups around them [12], and social support may be extremely powerful in motivating behavioral changes of individuals, families and organizations [13]. Therefore, theoretical models from behavioral sciences should be applied to better understand the determinants of adequate hand hygiene [9].

Rural Tibetan areas in China are characterized by a high incidence of foodborne parasitic diseases, especially echinococcosis [14]. Studies have shown that a major risk factor associated with echinococcosis is poor handwashing behavior, next to other factors such as living in a rural area, low income, owning dogs, and playing with dogs [15,16,17,18,19,20]. Adequate handwashing was suggested as a low-cost and efficient intervention against echinococcosis infections [2,3,4,5,6]. Primary school children living in rural risk areas may have poor hand hygiene habits and may be susceptible to foodborne parasitic diseases [21,22]. However, few studies have explored the reasons for poor hand hygiene among primary school students, especially those living in high foodborne parasitic disease (e.g., echinococcosis) incidence areas. Therefore, from the perspective of disease prevention, this study focuses on the determinants of handwashing of Tibetan primary school children using a comprehensive behavioral theory.

The specific study questions are (1) what are the handwashing habits of primary school children in rural Tibetan primary schools, and (2) what are the correlates of school children’s handwashing. The theory of reasoned action was applied in this study to explore the social and behavioral determinants that are associated with handwashing among children, as recommended by Gellman [23]. The theory of reasoned action entails that individual behavior is not only affected by individual’s own cognition, but also by his/her social relations [23]. The theory focuses on understanding the relationships between attitudes, subjective norms, behavioral intentions, and behaviors. Attitudes are, according to this theory, determined by individuals’ beliefs about outcomes or attributes of performing the behavior. Subjective norms are defined as a persons’ perception of and compliance to important referent individuals’ approval or disapproval of a particular behavior. The theory predicts that attitudes and subjective norms can predict an individual’s intention to carry out certain behavior [24]. Regarding poor hand hygiene of primary school students, this study aims to explore the correlates of school children’s handwashing from the perspective of the theory of reasoned action.

Based on the theory of reasoned action, our hypotheses are: (1) The knowledge and attitudes of students are associated with adequate hand hygiene; (2) the subjective norms, including the normative belief concerning important referent individuals and the motivation to comply with them, are associated with adequate hand hygiene; (3) families and peers, as well as teachers, play a role regarding the intention of students to wash their hands, and regarding adequate hand hygiene of the students. Therefore, the following model in Figure 1 is proposed.

## 2. Materials and Methods 

The study was reviewed and approved by the Institutional Review Board of the Western China School of Public Health, Sichuan University (project identification code: 0040405502146). Formal permission was obtained from the local Education Bureau of the Tibetan autonomous prefecture of Golog, and initial written consent was obtained from school principals, who were the students’ statutory guardians when they were boarding at school. Most students’ home was far away from the schools in Tibetan areas, and they could not go home often. Therefore, informed consent was obtained from school principals instead of parents for students’ participation in this study.

### 2.1. Design, Setting and Participants

The study design was a cross-sectional survey. The survey consisted of a one-time paper and pencil questionnaire to be completed by the students and was conducted in the Tibetan autonomous prefecture of Golog, Qinghai Province, one of the poorest areas in China [24]. Living in the Qinghai-Tibet Plateau with altitudes ranging from 3600 to 4500 meters, people in the region Golog face various challenges including slow social development and limited access to information [14].

Golog has a high prevalence (1.91%–11.93%) of human echinococcosis. Echinococcosis is a foodborne parasitic disease that can largely be prevented by adequate handwashing [14,25]. It is representative of characteristics of ethnically diverse regions, the plateau regions, and poverty-stricken areas in Western China [26].

In 2016, a multistage random cluster sampling method was applied to select the sample. Firstly, we randomly selected two of the six counties in Golog. Secondly, all nineteen townships from the sample counties were involved. Thirdly, we decided to include all the nineteen central primary schools in the nineteen townships, since there was only one central primary school in each township in the study area. Lastly, we purposively invited all students from the fourth to sixth grades to participate in our study; therefore, the students’ literacy level and language ability were adequate to understand and answer the questions.

### 2.2. Measures

Informed by the theory of reasoned action, and given the characteristics of the most important condition to be prevented in the study area, the questionnaire was designed based on a review of the literature and expert consultation. The questionnaire included the knowledge about preventing echinococcosis through handwashing, attitudes towards handwashing, and subjective norms including the normative belief concerning important referent individuals and the motivation to comply with them (see Table 2 and Figure 2 for an overview of the items). The structured questionnaire was internally consistent (Cronbach’s alpha = 0.85), and the construct validity was demonstrated by the accumulated variance contribution rate of 60.5%. The questionnaires were completed in the classrooms, allowing enough privacy for each participant; an investigator and a volunteer were present in each class to answer any question by the participants.

#### 2.2.1. Measure of Knowledge

Five items measured the knowledge of echinococcosis and handwashing. For instance, one item asked “can people be infected with echinococcosis if they don’t wash their hands before eating or after using the toilet?” A five-point Likert scale ranging from 1 (totally impossible) to 5 (totally possible) was utilized to measure participants’ responses to such knowledge based questions.

#### 2.2.2. Measure of Attitudes

Attitude towards handwashing was measured by three items such as “do you agree you should wash your hands before eating and after using the toilet?” A five-point Likert scale ranging from 1 (totally disagree) to 5 (totally agree) was utilized to measure these items. The higher the score, the more positive the attitude towards handwashing.

#### 2.2.3. Measure of Subjective Norms

The subjective norms were measured by the degree to which individuals were likely to comply with advice from three important referent individuals: family, teacher, and peer. Six pairs of items were included, such as “did your family ever tell you to wash hands before eating and after using the toilet?” followed by “to what extent are you likely to follow your family when they tell you to wash hands before eating and after using the toilet?” The first item on normative belief was measured by 0 (never told) and 1 (told). The second item on motivation to comply was measured by a five-point Likert scale ranging from 1 (totally impossible) to 5 (totally possible). The higher the total score, the more positive the subjective norm one perceived.

#### 2.2.4. Measure of Handwashing Behavior

The outcome variable is actual behavior since it is still unclear whether behavior intention could lead to actual behavior in the theory of reasoned action [27]. Four questions on handwashing behavior in daily life were included, such as “do you wash your hands before eating at home?” A five-point Likert scale ranging from 1 (never) to 5 (every time) was utilized to measure these items.

### 2.3. Statistical Analysis

We first examined sample demographic characteristics of participants and the rate of handwashing using descriptive statistics. We conducted confirmatory factor analysis to assess whether observed variables were adequate indicators of latent variables. Figure 1 shows the details of the hypothetical model. Structural equation model (SEM) was employed to further test above hypothesized relationship among knowledge, subjective norms, attitude, and handwashing in the model.

Several indicators were used to evaluate the fit of the model, goodness-of-fit index (GFI), comparative fit index (CFI), normed fit index (NFI) of 0.90 or above, χ^2^/degrees of freedom (χ^2^/df) less than 5.00, and the root mean square error of approximation (RMSEA) less than 0.05 [28]. The analyses above were performed using STATA14.2 (StataCorp, College Station, TX, United States) and AMOS 21.0 (IBM, Armonk, NY, United States) statistical software. The association was considered to be statistically significant if the bilateral test level was less than 0.05.

## 3. Results

### 3.1. Characteristics of the Population for Analysis

Table 1 shows the demographic characteristics of 1690 students. The average age was 11.3 years (SD = 1.6); 53.7% of respondents were female; 95.6% of students had siblings. Of the students in the study population, 35.0% were in grade 4, 32.8% in grade 5, and 32.2% in grade 6. Table A1 includes detailed information on numbers of student participants by school and grade. The educational level of 84.7% of the fathers and 76.6% of the mothers was primary school or below. On-site observations revealed that all schools in this study had toilets (pit latrine) as well as tap water and soap for handwashing. The pupil to latrine ratios ranged from 1/75 to 1/40 in the participating schools.

### 3.2. Descriptive Analysis of the Variables Utilized in the SEM

Table 2 shows the operational definitions and descriptive statistics of the following variables utilized in the SEM: knowledge, attitude, subjective norms, and handwashing. Most of the students (72.8%) had heard of echinococcosis, but only 44.1% knew echinococcosis is a kind of zoonosis (a disease that is transmitted to humans from animals). The susceptibility score of without washing hands was 3.70 (SD = 1.58). Nearly half of the participants knew echinococcosis could be prevented, and 42.8% of students knew echinococcosis could be prevented by washing hands frequently. The mean score of the attitudes towards ‘washing hands before meals and after toilet’ was 4.22 (SD = 1.10); the mean score of the attitude towards ‘washing hands after touching dogs’ was 4.29 (SD = 1.05); and the mean score of ‘washing hands in order to prevent echinococcosis’ was 4.16 (SD = 1.12). Regarding the subjective norms of washing hands before a meal and after using the toilet, as well as after touching dogs, participants most likely complied to suggestions from their teachers (4.22 ± 1.35, 4.12 ± 1.50), followed by their families (3.28 ± 2.09, 3.44 ± 2.03), and then peers (2.53 ± 2.27 and 2.39 ± 2.29). Lower percentages of students reported washing hands often and every time before meals at home (27.2%) than in school (38.1%).

### 3.3. Results of Structural Equation Model

The SEM was established to reflect the relationships pertaining to the variables in the research model. We fitted the data and the theoretical model through the maximum likelihood and bias-corrected bootstrap with 2000 replications, and modified the theoretical model until the AMOS reached a local minimum according to model fit indices. The model fitness was within the acceptable ranges, namely, the normed chi–square (χ^2^/df = 4.432), goodness-of-fit index (GFI = 0.965), comparative fit index (CFI = 0.952), normed fit index (NFI = 0.961) as well as root mean square error of approximation (RMSEA = 0.045). Figure 2 is the diagrammatic representation of the results of the fitted model together with the standardized estimation of path regression weights or path coefficients (β).

The total effects, direct effects and indirect effects among the four latent variables were shown in Table 3. Attitude (β = 0.22, 95% CI 0.13–0.31) and subjective norm (β = 0.09, 95% CI 0.01–0.18) affected students’ handwashing behavior directly. Therefore, the result supports the first hypothesis that attitude and subjective norms directly affect handwashing, and the subjective norm has an indirect effect on students’ handwashing behavior through attitude. The path coefficient value was β = 0.12 (95% CI 0.07–0.17), so the total effect value that subjective norm affected students’ handwashing behavior was β = 0.21 (95% CI 0.14–0.27). Moreover, knowledge had an indirect effect on students’ handwashing behavior not only via attitude, but also via subjective norm, and the path coefficient value of subjective norm was β = 0.04 (95% CI 0.03–0.07). Regarding the second hypothesis, Figure 2 illustrates teachers have a relatively higher association with the students’ subjective norms in washing hands before meals and after using the toilet (β = 0.74, *p* < 0.001), as well as washing hands after touching dogs (β = 0.76, *p* < 0.001) compared to families and peers. This result supports our hypothesis that teachers have a relatively stronger social influence on primary school students’ handwashing compliance than families and peers. Primary school students may be more compliant to their teachers than to their families and peers regarding handwashing.

## 4. Discussion

In the current study, we constructed a structural equation model based on the theory of reasoned action to explore the correlates of primary school students’ handwashing behavior. All hypotheses were supported by our results. The results supported the first hypothesis that subjective norms and attitude could directly affect handwashing. Furthermore, knowledge and subjective norms indirectly affect handwashing via attitude [29]. Lastly, the normative belief concerning important referent individuals and the motivation to comply with them contributed to hand hygiene habits, and teachers were found to have greater influence on students’ subjective norms than families and peers.

The results showed a small percentage of students frequently wash hands before meals and after toilet at home or school, and this percentage is lower than the findings of other studies in adjacent regions [12,30]. Several studies have established the relationship between handwashing and a decrease in parasitic infection among school-aged students in rural areas with high endemic infection rates [27,29]. In remote rural areas in China, most of the primary schools are boarding schools, and students live in the school on weekdays [21,31]. If students do not have a good habit of handwashing before meals and after using the toilet, their hands could be contaminated with parasitic eggs, and their peers might be infected when they play together. Previous research proved that close contact between students with bad hand hygiene practices was one of the main causes of the spread of fecal-oral transmission disease within school [32]. Our study provides empirical evidence regarding the factors that influence students’ hand hygiene practice.

The results of the structural equation model showed that attitudes to handwashing had a strong direct association with primary school students’ handwashing behavior. Students who had favorable attitudes to handwashing were more likely to wash their hands, which is consistent with findings from a previous study based on the RANAS (risk, attitudes, norms, abilities, self–regulation) model [33]. In this study, knowledge had a small indirect effect on handwashing via attitude. Previous studies have shown many other factors that influence sustained handwashing behavior, such as having time to wash hands using accessible, clean facilities, being encouraged through the existence of handwashing opportunities in the daily routine, and handwashing being viewed as the social norm [2,34]. Findings from this study confirm the existing literature that knowledge and attitudes are factors that may influence behavior, especially the effect of attitudes [10]. A health intervention study suggested that the attitude towards health aspects of sanitation behavior is important because it determines the degree of sustainability of hygiene promotion in sanitation [10]. Poor knowledge and practice of, as well as negative attitudes to personal hygiene have negative consequences for a child’s long-term overall development [35]. Therefore, it is necessary to cultivate students’ positive attitudes towards handwashing in primary schools, so that they can realize the benefits of frequent handwashing, especially in echinococcosis endemic areas.

The present study also found that subjective norms not only had the direct positive effects on students’ handwashing behavior, but also had important indirect effects via attitude. This finding indicated that subjective norms played an important role in students’ handwashing behavior, which is consistent with previous research findings [36,37]. Thus, our study suggests future intervention programs to incorporate strategies that strengthen the normative belief concerning significant others and the motivation to comply with these significant others, with the purpose of reinforcing students’ attitude towards and practice of handwashing. The evaluation of those programs could provide further empirical evidence on the applicability of our findings on the subjective norms’ direct and indirect positive influences on students’ handwashing behavior.

An important new finding of this study is that teachers had a stronger influence on students’ handwashing behavior, compared with families and peers. In remote rural areas in China, most primary students board in their schools and spend more time with their teachers than parents except during vacation time [21,32]. Teachers are not only responsible for imparting knowledge to students, but also guiding students’ daily behaviors, including sanitary habits [38]. Moreover, teachers are highly respected and trusted by primary school students. Existing research suggests teachers’ adequate support for students is the key to implement and sustain hand hygiene programs among students [39]. Our finding that students have a high compliance with the normative beliefs of teachers is in line with this. This supports the importance of school-based interventions.

Findings of this study provide important information for health care practitioners, families, and educators to design educational and intervention programs to improve hand hygiene among students. Given the setting of the study, the results may be especially relevant for the development of interventions in economically disadvantaged areas.

Several limitations of this study need to be taken into consideration. First, because of the cross-sectional design, we cannot infer causal relationships between the factors that we identified and the handwashing behaviors. Second, social desirability may have affected the answers given by the participants. This study was conducted in remote and poor areas. Therefore, we recommend replicating this study in other populations and regions. We also suggest future studies to apply a longitudinal design and to validate the measures of handwashing behavior by observations.

The findings of this study may be used to develop interventions to promote adequate handwashing among school children. When evaluating such interventions, it is important not only to evaluate the handwashing behaviors, but also to evaluate whether the intervention and the handwashing indeed lower the risk for the infectious conditions that are targeted by the intervention in a specific context or region [2]. So, in future studies, parasitological methods are recommended to be applied to evaluate the impacts on the transmission of the soil-transmitted parasitic infections (i.e., alveolar echinococcosis, cystic echinococcosis, etc.) in each specific context or region [40].

In this study, regarding the measurement of the attitude towards ‘handwashing before meals and after toilet’, it should be considered that the attitude towards these two elements (i.e., ‘handwashing before meals’, and ‘handwashing after toilet use’) can vary within one person. Therefore, we propose future studies to measure these two attitudes separately.

Finally, the theory of reasoned action model was applied, which does not include perceived threat as a determinant of behavior. Perceived threat contains ‘severity’ and ‘perceived susceptibility’ [41]. Yang found that ‘perceived severity’ is associated with handwashing behavior, while ‘perceived susceptibility’ is not [26]. Therefore, we additionally explored whether ‘perceived threat’ was associated with handwashing and added this variable to our model. However, we found no statistically significant associations between ‘perceived threat’ and handwashing behavior. We recommend future studies further examine the role of perceived threat in handwashing behavior among students.

## 5. Conclusions

This study found that the students in this study had poor handwashing behaviors before meals and after toilet both at home and in school in echinococcosis endemic areas. Students’ handwashing behaviors were directly associated with attitudes, and subjective norms affected handwashing behaviors both directly and indirectly. Teachers had a stronger influence on students’ subjective norms compared to families and peers. Findings from this study suggest that teacher-involved participatory hygiene education could be an effective way to promote students’ handwashing behaviors in echinococcosis endemic areas.

## Figures and Tables

**Figure 1 ijerph-16-03217-f001:**
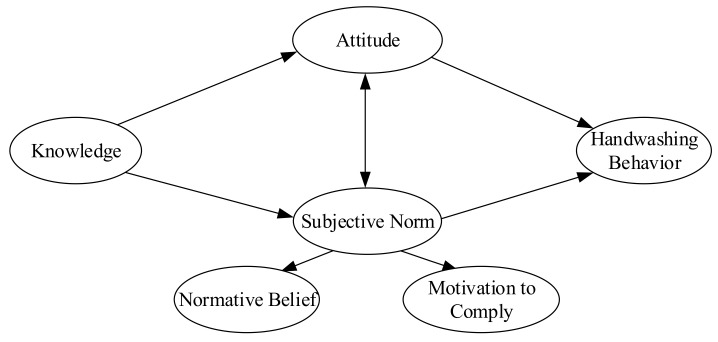
Hypothetical model.

**Figure 2 ijerph-16-03217-f002:**
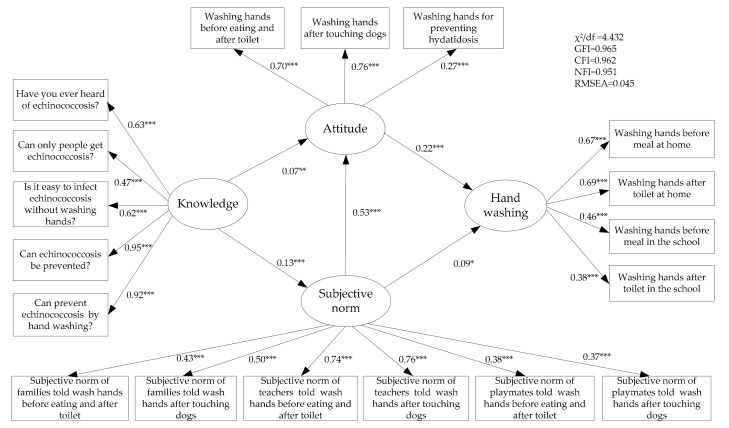
Standardized path coefficients and the relationships of the variables. Note: Values indicate standardized coefficients; * *p* < 0.05, ** *p* < 0.01, *** *p* < 0.001.

**Table 1 ijerph-16-03217-t001:** Demographic characteristics of the study population (N = 1690).

Demographic Variables		n/Mean ± SD	(%)
Age		11.27 ± 1.58	
Gender	Male	782	46.3
	Female	908	53.7
Grade	4	592	35.0
	5	554	32.8
	6	544	32.2
Only child	Yes	75	4.44
	No	1615	95.6
Mother’s education	Primary school or below	1431	84.7
	Middle school	176	10.4
	High school or more	83	4.91
Father’s education	Primary school or below	1294	76.6
	Middle school	274	16.2
	High school or more	122	7.22

**Table 2 ijerph-16-03217-t002:** Descriptive statistics for the variables utilized in the SEM (N = 1690).

Latent Variables	Observed Variables	n (%)/Mean ± SD
Knowledge	Have you ever heard of echinococcosis (yes)	1231 (72.8%)
	What disease is echinococcosis (correct)	745 (44.1%)
	Susceptibility of echinococcosis without washing hands (1–5) ^1^	3.70 ± 1.58
	Echinococcosis can be prevented (yes)	834 (49.4%)
	Frequent hands washing can prevent echinococcosis (yes)	723 (42.8%)
Attitude ^2^	Washing hands before eating and after toilet (1–5) ^2^	4.22 ± 1.10
	Washing hands after touching dogs (1–5) ^2^	4.29 ± 1.05
	Washing hands for preventing echinococcosis (1–5) ^2^	4.16 ± 1.12
Subjective norm ^3^	Following family’s suggestion to wash hands before meal and after toilet (0–5) ^3^	3.28 ± 2.09
	Following family’s suggestion to wash hands after touching dogs (0–5) ^3^	3.44 ± 2.03
	Following teachers’ suggestion to wash hands before meal and after toilet (0–5) ^3^	4.22 ± 1.35
	Following teachers’ suggestion to wash hands after touching dogs (0–5) ^3^	4.12 ± 1.50
	Following peers’ suggestion to wash hands before meal and after toilet (0–5) ^3^	2.53 ± 2.27
	Following peers’ suggestion to wash hands after touching dogs (0–5) ^3^	2.39 ± 2.29
Handwashing	Washing hands before meal at home (1–5) ^4^	2.92 ± 1.17
	Never	192 (11.4%)
	Rarely	421 (24.9%)
	Seldom	617 (36.5%)
	Often	242 (14.3%)
	Every time	218 (12.9%)
	Washing hands after using toilet at home (1–5) ^4^	2.77 ± 1.31
	Never	343 (20.3%)
	Rarely	421 (24.9%)
	Seldom	460 (27.2%)
	Often	219 (13.0%)
	Every time	247 (14.6%)
	Washing hands before meal in the school (1–5) ^4^	3.09 ± 1.32
	Never	260 (15.4%)
	Rarely	297 (17.6%)
	Seldom	490 (29.0%)
	Often	315 (18.6%)
	Every time	328 (19.4%)
	Washing hands after using toilet in the school (1–5) ^4^	2.64 ± 1.28
	Never	401 (23.7%)
	Rarely	390 (23.1%)
	Seldom	484 (28.6%)
	Often	231 (13.7%)
	Every time	184 (10.9%)

Note: ^1^ A five-point Likert scale ranging from 1 (totally impossible) to 5 (totally possible) was utilized to measure this item. ^2^ Attitude is degree of agreement on the listed items. A five-point Likert scale ranging from 1 (totally disagree) to 5 (totally agree) was utilized to measure these items. ^3^ Subjective norm was multiplying normative beliefs by the motivation to comply with them. The higher the total score after multiplying, the more positive the subjective norm one perceived. ^4^ A five-point Likert scale ranging from 1 (never) to 5 (every time) was utilized to measure this item.

**Table 3 ijerph-16-03217-t003:** Results of the structural equation modeling.

Model Pathways	Total Effects	Direct Effects	Indirect Effects
Knowledge → Handwashing	0.04 (0.03–0.07)	–––	0.04 (0.01–0.07)
Subjective norm → Handwashing	0.21 (0.14–0.28)	0.09 (0.01–0.18)	0.12 (0.07–0.17)
Attitude → Handwashing	0.22 (0.13–0.31)	0.22 (0.13–0.31)	–––

Note: Values indicate standardized coefficients and 95% CI.

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
