# Peer review of "Correlates of School Children’s Handwashing: A Study in Tibetan Primary Schools"

_ijerph, 2019, doi:10.3390/ijerph16173217_

Round 1

Reviewer 1 Report

I have no further comments or suggestions.

Reviewer 2 Report

Thank you for addressing the comments from the previous review so thoroughly. The additional content and references have, in my opinion, improved the quality and the readability of the manuscript and made it acceptable for publication. 

This manuscript is a resubmission of an earlier submission. The following is a list of the peer review reports and author responses from that submission.

Round 1

Reviewer 1 Report

The study by Sun et al. verified the factors related to handwashing in school children in an area located in the Tibetan autonomous prefecture of Golog. Regarding the study about handwashing, it is well conducted and with sound results. However, the study did not perform any parasitological method in order to really make a correlation with infection and handwashing as a preventive attitude.

The English language is confusing in several parts of the MS and a careful review is recommended. There are several orthographical errors that must be corrected.

In the introduction, the authors cite a study on taeniasis/cysticercosis, misleading the readers (line 60).  

Table 1, only "female" appears to be a gender variable.

A more careful description of the results is welcome eg. (line178) there is no explanation of the numbers described for the attitude of washing hands.

An important point is the lack of any result related with echinococcosis in the studied population, without this information it is speculative to correlate the attitude of handwashing with prevention of echinococcosis.

In my opinion, this MS could focus on handwashing only without stress the prevention of echinococcosis.

Reviewer 2 Report

I enjoyed your manuscript and the use of echinococcosis as a platform for examining student hand washing practices and behavioural factors. Overall the manuscript flows well and conclusions are supported by the results and modelling. I have minor comments in relation to the content for you to take under consideration. 

General comments 

Perhaps a limitation of the study or the data presented here is that you have not explored how respondents 'feel' about being infected with Echinococcus? Was this done and not presented? If you are to use this infection through the basis of attitudes and norms for promoting hand hygiene at critical times then it would be important to know if it is perceived as a significant illness or threat. 

There is reference to dogs as a significant risk throughout the paper but I am assuming that there are no dogs at the school? This should be clearly stated. 

There have been recent reviews on hand washing in schools, behaviour change and the use of environmental prompts to support improve hand washing which I would expect to see referred to in this manuscript and would recommend the authors refer to during their review process. 

        e.g. Elise Grover (Bangladesh), and https://www.mdpi.com/1660-4601/16/3/359   

Content

Check text to ensure it refers the methodology and results using past tense

References 23 could be replaced with more relevant recent studies and reviews which have been specific to school children rather than college students. 

L105 provide a reference to support the commentary made on the study area

L107 provide value for the prevalence of Echinococcus in the area

Section 2.2.1 - I have concerns that hand washing after eating and after toilet were put together rather than explored separately in terms of attitudes. This decision needs to be justified as the reasons for doing each could be completely different. 

Section 3.1 - I would like to see more detail on how many students from each school and what percentage of students from each year group participated so that representation can be determined. 

L163 - the line describing the percentage of students from each year group is confusing at the moment and can be simplified to make the meaning more clear. 

L165 - reference to parent education level values in brackets is confusing and only became clear once I referred to the table indicating mother versus father. This should be clarified.

L166 - it is not clear to the reader what is meant by 'sufficient facilities' and this should be provided in more detail so that the reader can determine the type of toilets, ratio to students, type of water source for hand washing, availability of soap, etc. Without this information the reader cannot determine if there were challenges with ability factors to achieve hand washing.

L250 infers that if the students have knowledge then they will wash their hands. However we know that achieving sustained hand washing in any population is much more complex than this and has to take issues of behaviour change into consideration. This section should be revisited and updated with appropriate references to ensure it reflects currently research and understanding of this process.  

Text edits

L50 'were' should be 'are'

L54 Tibetan areas 'are' strongly

L66 replace 'of' with 'with'

L69 after 'research' add 'studies have'

L101 replace 'is' with 'was'

L123 replace 'measure' with 'measured'

L124 replace 'asks' with 'asked'

L159 change 'is' to 'was'

L129 replace 'influences' with 'influence'

L238 insert after 'using the' toilet

Round 2

Reviewer 1 Report

The weak point of this MS is still the lack of cause-effect correlation is the study. Once there is no proof the students with washing hands behavior presented less worm burden (or any other parasitological status), the objectives were not reached. 

I recommend to rewrite the MS 1)adding the parasitological status of the population pre and post handwashing intervention,   or 2)find any other parameter that changed after handwashing.